# NEURAL SEPARATION OF OBSERVED AND UNOBSERVED DISTRIBUTIONS

## ABSTRACT

Separating mixed distributions is a long standing challenge for machine learning and signal processing. Applications include: single-channel multi-speaker separation (cocktail party problem), singing voice separation and separating reflections from images. Most current methods either rely on making strong assumptions on the source distributions (e.g. sparsity, low rank, repetitiveness) or rely on having training samples of each source in the mixture. In this work, we tackle the scenario of extracting an unobserved distribution additively mixed with a signal from an observed (arbitrary) distribution. We introduce a new method: Neural Egg Separation - an iterative method that learns to separate the known distribution from progressively finer estimates of the unknown distribution. In some settings, Neural Egg Separation is initialization sensitive, we therefore introduce GLO Masking which ensures a good initialization. Extensive experiments show that our method outperforms current methods that use the same level of supervision and often achieves similar performance to full supervision.

## 1 INTRODUCTION

Humans are remarkably good at separating data coming from a mixture of distributions, e.g. hearing a person speaking in a crowded cocktail party. Artificial intelligence, on the the hand, is far less adept at separating mixed signals. This is an important ability as signals in nature are typically mixed, e.g. speakers are often mixed with other speakers or environmental sounds, objects in images are typically seen along other objects as well as the background. Understanding mixed signals is harder than understanding pure sources, making source separation an important research topic.

Mixed signal separation appears in many scenarios corresponding to different degrees of supervision. Most previous work focused on the following settings:

*Full supervision:* The learner has access to a training set including samples of mixed signals $\{y_i\} \in \mathcal{Y}$ as well as the ground truth sources of the same signals $\{b_i\} \in \mathcal{B}$ and $\{x_i\} \in \mathcal{X}$ (such that $y_i = x_i + b_i$). Having such strong supervision is very potent, allowing the learner to directly learn a mapping from the mixed signal $y_i$ to its sources $(x_i, b_i)$. Obtaining such strong supervision is typically unrealistic, as it requires manual separation of mixed signals. Consider for example a musical performance, humans are often able to separate out the different sounds of the individual instruments, despite never having heard them play in isolation. The fully supervised setting does not allow the clean extraction of signals that cannot be observed in isolation e.g. music of a street performer, car engine noises or reflections in shop windows.

*Synthetic full supervision:* The learner has access to a training set containing samples from the mixed signal $\{y_i\} \in \mathcal{Y}$ as well as samples from all source distributions $\{b_j\} \in \mathcal{B}$ and $\{x_k\} \in \mathcal{X}$. The learner however does not have access to paired sets of the mixed and unmixed signal ground truth (that is for any given $y_i$ in the training set, $b_i$ and $x_i$ are unknown). This supervision setting is more realistic than the fully supervised case, and occurs when each of the source distributions can be sampled in its pure form (e.g. we can record a violin and piano separately in a studio and can thus obtain unmixed samples of each of their distributions). It is typically solved by learning to separate synthetic mixtures $b_j + x_k$ of randomly sampled $b_j$ and $x_k$.

*No supervision:* The learner only has access to training samples of the mixed signal $\mathcal{Y}$ but not to sources $\mathcal{B}$ and $\mathcal{X}$. Although this settings puts the least requirements on the training dataset, it is

a hard problem and can be poorly specified in the absence of strong assumptions and priors. It is generally necessary to make strong assumptions on the properties of the component signals (e.g. smoothness, low rank, periodicity) in order to make progress in separation. This unfortunately severely limits the applicability of such methods.

In this work we concentrate on the *semi-supervised* setting: unmixing of signals in the case where the mixture $\mathcal{Y}$ consists of a signal coming from an unobserved distribution $\mathcal{X}$ and another signal from an observed distribution $\mathcal{B}$ (i.e. the learner has access to a training set of clean samples such that $\{b_j\} \in \mathcal{B}$ along with different mixed samples $\{y_i\} \in \mathcal{Y}$). One possible way of obtaining such supervision, is to label every signal sample by a label, indicating if the sample comes only from the observed distribution $\mathcal{B}$ or if it is a mixture of both distributions $\mathcal{B} + \mathcal{X}$. The task is to learn a parametric function able to separate the mixed signal $y_i \in \mathcal{Y}$ into sources $x_i \in \mathcal{X}$ and $b_i \in \mathcal{B}$ s.t. $y_i = b_i + x_i$. Such supervision is much more generally available than full supervision, while the separation problem becomes much simpler than when fully unsupervised.

We introduce a novel method: Neural Egg Separation (NES) - consisting of $i$) iterative estimation of samples from the unobserved distribution $\mathcal{X}$ $ii$) synthesis of mixed signals from known samples of $\mathcal{B}$ and estimated samples of $\mathcal{X}$ $iii$) training of separation regressor to separate the mixed signal. Iterative refinement of the estimated samples of $\mathcal{X}$ significantly increases the accuracy of the learned masking function. As an iterative technique, NES can be initialization sensitive. We therefore introduce another method - GLO Masking (GLOM) - to provide NES with a strong initialization. Our method trains two deep generators end-to-end using GLO to model the observed and unobserved sources ($\mathcal{B}$ and $\mathcal{X}$). NES is very effective when $\mathcal{X}$ and $\mathcal{B}$ are uncorrelated, whereas initialization by GLOM is most important when $\mathcal{X}$ and $\mathcal{B}$ are strongly correlated such as e.g. separation of musical instruments. Initialization by GLOM was found to be much more effective than by adversarial methods.

Experiments are conducted across multiple domains (image, music, voice) validating the effectiveness of our method, and its superiority over current methods that use the same level of supervision. Our semi-supervised method is often competitive with the fully supervised baseline. It makes few assumptions on the nature of the component signals and requires lightweight supervision.

## 2 PREVIOUS WORK

*Source separation:* Separation of mixed signals has been extensively researched. In this work, we focus on single channel separation. Unsupervised (blind) single-channel methods include: ICA (Davies & James, 2007) and RPCA (Huang et al., 2012). These methods attempt to use coarse priors about the signals such as low rank, sparsity or non-gaussianity. HMM can be used as a temporal prior for longer clips (Roweis, 2001), however here we do not assume long clips. Supervised source separation has also been extensively researched, classic techniques often used learned dictionaries for each source e.g. NMF (Wilson et al., 2008). Recently, neural network-based gained popularity, usually learning a regression between the mixed and unmixed signals either directly (Huang et al., 2014) or by regressing the mask (Wang et al., 2014; Yu et al., 2017). Some methods were devised to exploit the temporal nature of long audio signal by using RNNs (Mimilakis et al., 2017), in this work we concentrate on separation of short audio clips and consider such line of works as orthogonal. One related direction is Generative Adversarial Source Separation (Stoller et al., 2017; Subakan & Smaragdis, 2017) that uses adversarial training to match the unmixed source distributions. This is needed to deal with correlated sources for which learning a regressor on synthetic mixtures is less effective. We present an Adversarial Masking (AM) method that tackles the semi-supervised rather than the fully supervised scenario and overcomes mixture collapse issues not present in the fully supervised case. We found that non-adversarial methods perform better for the initialization task.

The most related set of works is semi-supervised audio source separation (Smaragdis et al., 2007; Barker & Virtanen, 2014), which like our work attempt to separate mixtures $\mathcal{Y}$ given only samples from the distribution of one source $\mathcal{B}$. Typically NMF or PLCA (which is a similar algorithm with a probabilistic formulation) are used. We show experimentally that our method significantly outperforms NMF.

*Disentanglement:* Similarly to source separation, disentanglement also deals with separation in terms of creating a disentangled representation of a source signal, however its aim is to uncover

latent factors of variation in the signal such as style and content or shape and color e.g. Denton et al. (2017); Higgins et al. (2016). Differently from disentanglement, our task is separating signals rather than the latent representation.

*Generative Models:* Generative models learn the distribution of a signal directly. Classical approaches include: SVD for general signals and NMF (Lee & Seung, 2001) for non-negative signals. Recently several deep learning approaches dominated generative modeling including: GAN (Goodfellow et al., 2016), VAE (Kingma & Welling, 2013) and GLO (Bojanowski et al., 2018). Adversarial training (for GANs) is rather tricky and often leads to mode-collapse. GLO is non-adversarial and allows for direct latent optimization for each source making it more suitable than VAE and GAN.

## 3 NEURAL EGG SEPARATION (NES)

In this section we present our method for separating a mixture of sources of known and unknown distributions. We denote the mixture samples $y_i$, the samples with the observed distribution $b_i$ and the samples from the unobserved distribution $x_i$. Our objective is to learn a parametric function $T()$, such that $b_i = T(y_i)$.

**Full Supervision:** In the fully supervised setting (where pairs of $y_i$ and $b_i$ are available) this task reduces to a standard supervised regression problem, in which a parametric function $T()$ (typically a deep neural network) is used to directly optimize:

$$\arg\min_T L = \sum_i \ell(T(y_i), b_i) \tag{1}$$

Where typically $\ell$ is the Euclidean or the $L_1$ loss. In this work we use $\ell() = L_1()$.

Mixed-unmixed pairs are usually unavailable, but in some cases it is possible to obtain a training set which includes unrelated samples $x_j$ and $b_k$ e.g. (Wang et al., 2014; Yu et al., 2017). Methods typically randomly sample $x_j$ and $b_k$ sources and synthetically create mixtures $y_{jk} = x_j + b_k$. The synthetic pairs $(b_k, y_{jk})$ can then be used to optimize Eq. 1. Note that in cases where $\mathcal{X}$ and $\mathcal{B}$ are correlated (e.g. vocals and instrumental accompaniment which are temporally dependent), random synthetic mixtures of $x$ and $b$ might not be representative of $y$ and fail to generalize on real mixtures.

**Semi-Supervision:** In many scenarios, clean samples of both mixture components are not available. Consider for example a street musical performance. Although crowd noises without street performers can be easily observed, street music without crowd noises are much harder to come by. In this case therefore samples from the distribution of crowd noise $\mathcal{B}$ are available, whereas the samples from the distribution of the music $\mathcal{X}$ are unobserved. Samples from the distribution of the mixed signal $\mathcal{Y}$ i.e. the crowd noise mixed with the musical performance are also available.

The example above illustrates a class of problems for which the distribution of the mixture and a single source are available, but the distribution of another source is unknown. In such cases, it is not possible to optimize Eq. 1 directly due to the unavailability of pairs of $b$ and $y$.

**Neural Egg Separation:** Fully-supervised optimization (as in Eq. 1) is very effective when pairs of $b_i$ and $y_i$ are available. We present a novel algorithm, which iteratively solves the semi-supervised task as a sequence of supervised problems without any clean training examples of $\mathcal{X}$. We name the method Neural Egg Separation (NES), as it is akin to the technique commonly used for separating egg whites and yolks.

The core idea of our method is that although no clean samples from $\mathcal{X}$ are given, it is still possible to learn to separate mixtures of observed samples $b_j$ from distribution $\mathcal{B}$ combined with some estimates of the unobserved distribution samples $\tilde{x}_i$. Synthetic mixtures are created by randomly sampling an approximate sample $\tilde{x}_i$ from the unobserved distribution and combining with training sample $b_j$:

$$\tilde{y}_{ij} = b_j + \tilde{x}_i \tag{2}$$

thereby creating pairs $(\tilde{y}_{ij}, b_j)$ for supervised training. Note that the distribution of synthetic mixtures $\tilde{y}_{ij}$ might be different from the real mixture sample distribution $y_j$, but the assumption (which is empirically validated) is that it will eventually converge to the correct distribution.

During each iteration of NES, a neural separation function $T()$ is trained on the created pairs by optimizing the following term:

$$T = arg \min_{T} L_1(T(b_j + \tilde{x}_i), b_j) \qquad (3)$$

At the end of each iteration, the separation function $T()$ can be used to approximately separate the training mixture samples $y_i$ into their sources:

$$\tilde{b}_i = T(y_i), \tilde{x}_i = y_i - \tilde{b}_i \qquad (4)$$

The refined $\mathcal{X}$ domain estimates $\tilde{x}_i$ are used for creating synthetic pairs for finetuning $T()$ in the next iteration (as in Eq. 3).

The above method relies on having an estimate of the unobserved distribution samples as input to the first iteration. One simple scheme is to initialize the estimates of the unobserved distribution samples in the first iteration as $\tilde{x}_i = c \cdot y_i$, where $c$ is a constant fraction (typically 0.5). Although this initialization is very naive, we show that it achieves very competitive performance in cases where the sources are independent. More advanced initializations will be discussed below.

At test time, separation is simply carried out by a single application of the trained separation function $T()$ (exactly as in Eq. 4).

**Data:** Mixture samples $\{y_i\}$, Observed source samples $\{b_j\}$
**Result:** Separation function $T()$
Initialize synthetic unobservable samples with $\tilde{x}_i \leftarrow c \cdot y_i$ or using AM or GLOM;
Initialize $T()$ with random weights;
**while** $iter < N$ **do**
    Synthesize mixtures $\tilde{y}_{ij} = b_j + \tilde{x}_i$ for all $b_j$ in $\mathcal{B}$;
    Optimize separation function for $P$ epochs: $T = arg \min_T \sum_i L_1(T(\tilde{y}_{ij}), b_i)$;
    Update estimates of unobserved distribution samples: $\tilde{x}_i = y_i - T(y_i)$
**end**

**Algorithm 1:** NES Algorithm

Our full algorithm is described in Alg. 1. For optimization, we use SGD using ADAM update with a learning rate of 0.01. In total we perform $N = 10$ iterations, each consisting of optimization of $T$ and estimation of $\tilde{x}_i$, $P = 25$ epochs are used for each optimization of Eq. 3.

**GLO Masking:** NES is very powerful in practice despite its apparent simplicity. There are some cases for which it can be improved upon. As with other synthetic mixture methods, it does not take into account correlation between $\mathcal{X}$ and $\mathcal{B}$ e.g. vocals and instrumental tracks are highly related, whereas randomly sampling pairs of vocals and instrumental tracks is likely to synthesize mixtures quite different from $\mathcal{Y}$. Another issue is finding a good initialization—this tends to affect performance more strongly when $\mathcal{X}$ and $\mathcal{B}$ are dependent.

We present our method GLO Masking (GLOM), which separates the mixture by a distributional constraint enforced via GLO generative modeling of the source signals. GLO (Bojanowski et al., 2018) learns a generator $G()$, which takes a latent code $z_b$ and attempts to reconstruct an image or a spectrogram: $b = G(z_b)$. In training, GLO learns end-to-end both the parameters of the generator $G()$ as well as a latent code $z_b$ for every training sample $b$. It trains per-sample latent codes by *direct gradient descent over the values of $z_b$* (similar to word embeddings), rather than by a feedforward encoder used by autoencoders (e.g. $z_b = E(b)$). This makes it particularly suitable for our scenario. Let us define the set of latent codes: $Z = [z_1..z_N]$. The optimization is therefore:

$$\arg \min_{Z,G} \sum_{b \in \mathcal{B}} \|b, G(z_b)\| \qquad (5)$$

We propose GLO Masking, which jointly trains generators: $G_B()$ for $\mathcal{B}$ and $G_X()$ for $\mathcal{X}$ such that their sum results in mixture samples $y = G_B(z_y^B) + G_X(z_y^X)$. We use the supervision of the observed source $\mathcal{B}$ to train $G_B()$, while the mixture $\mathcal{Y}$ contributes residuals that supervise the

training of $G_X()$. We also jointly train the latent codes for all training images: $z_b \in Z$ for all $b \in \mathcal{B}$, and $z_y^B \in Z^B, z_y^X \in Z^X$ for all $y \in \mathcal{Y}$. The optimization problem is:

$$\underset{Z,Z_B,Z_X,G_B,G_X}{\arg\min} \sum_{b \in \mathcal{B}} \|b, G_B(z_b)\| + \lambda \cdot \sum_{y \in \mathcal{Y}} \|y, G_B(z_y^B) + G_X(z_y^X)\| \qquad (6)$$

As GLO is able to overfit arbitrary distributions, it was found that constraining each latent code vector $z$ to lie within the unit ball $z \cdot z \leq 1$ is required for generalization. Eq. 6 can either be optimized end-to-end, or the left-hand term can be optimized first to yield $Z, G_B()$, then the right-hand term is optimized to yield $Z_B, Z_X, G_X()$. Both optimization procedures yield similar performance (but separate training does not require setting $\lambda$). Once $G_B()$ and $G_X()$ are trained, for a new mixture sample we infer its latent codes:

$$\underset{z_x,z_b}{\arg\min} \|y, G_B(z_y^B) + G_X(z_y^X)\| \qquad (7)$$

Our estimate for the sources is then:

$$\tilde{b} = G_B(z_y^B) \quad \tilde{x} = G_X(z_y^X) \qquad (8)$$

**Masking Function:** In separation problems, we can exploit the special properties of the task e.g. that the mixed signal $y_i$ is the sum of two positive signals $x_i$ and $b_i$. Instead of synthesizing the new sample, we can instead simply learn a separation mask $m()$, specifying the fraction of the signal which comes from $\mathcal{B}$. The attractive feature of the mask is always being in the range $[0, 1]$ (in the case of positive additive mixtures of signals). Even a constant mask will preserve all signal gradients (at the cost of introducing spurious gradients too). Mathematically this can be written as:

$$T(y) = y \odot m(y) \qquad (9)$$

For NES (and baseline AM described below), we implement the mapping function $T(y_i)$ using the product of the masking function $y_i \cdot m(y_i)$. In practice we find that learning a masking function yields much better results than synthesizing the signal directly (in line with other works e.g. Wang et al. (2014); Gabbay et al. (2017)).

GLOM models each source separately and is therefore unable to learn the mask directly. Instead we refine its estimate by computing an effective mask from the element-wise ratio of estimated sources:

$$m = \frac{G_B(z_b)}{G_B(z_b) + G_X(z_x)} \qquad (10)$$

**Initializing Neural Egg Separation by GLOM:** Due to the iterative nature of NES, it can be improved by a good initialization. We therefore devise the following method: $i$) Train GLOM on the training set and infer the mask for each mixture. This is operated on images or mel-scale spectrograms at $64 \times 64$ resolutions $ii$) For audio: upsample the mask to the resolution of the high-resolution linear spectrogram and compute an estimate of the $\mathcal{X}$ source linear spectrogram on the training set $iii$) Run NES on the observed $\mathcal{B}$ spectrograms and estimated $\mathcal{X}$ spectrograms. We find experimentally that this initialization scheme improves NES to the point of being competitive with fully-supervised training in most settings.

## 4  EXPERIMENTS

To evaluate the performance of our method, we conducted experiments on distributions taken from multiple real-world domains: images, speech and music, in cases where the two signals are correlated and uncorrelated.

We evaluated our method against 3 baseline methods:

*Constant Mask (Const):* This baseline uses the original mixture as the estimate.

*Semi-supervised Non-negative Matrix Factorization (SS-NMF):* This baseline method, proposed by Smaragdis et al. (2007), first trains a set of $l$ bases on the observed distribution samples $B$ by Sparse

Figure 1: A Qualitative Separation Comparison on Mixed Bag and Shoe Images

| Const | NMF | AM | GLOM | NES | FT | Sup | GT |
|---|---|---|---|---|---|---|---|

NMF (Hoyer, 2004; Kim & Park, 2007). It factorizes $B = H_b * W_b$, with activations $H_b$ and bases $W_b$, all matrices are non-negative. The optimization is solved using the Non-negative Least Squares solver by Kim & Park (2011). It then proceeds to train another factorization on the mixture $Y$ training samples with $2l$ bases, where the first $l$ bases ($W_b$) are fixed to those computed in the previous stage: $Y = H_{yb} * W_b + H_{yx} * W_x$. The separated sources are then: $\tilde{x} = h_{yx} * W_x$ and $\tilde{b} = h_{yb} * W_b$. More details can be found in the appendix B.

*Adversarial Masking (AM):* As an additional contribution, we introduce a new semi-supervised method based on adversarial training, to improve over the shallow NMF baseline. AM trains a masking function $m()$ so that after masking, the training mixtures are indistinguishable from the distribution of source $\mathcal{B}$ under an adversarial discriminator $D()$. The loss functions (using LS-GAN (Mao et al., 2017)) are given by:

$$\arg\min_D L_D = \sum_{y \in \mathcal{Y}} D(y \odot m(y))^2 + \sum_{b \in \mathcal{B}} (D(b) - 1)^2 \qquad \arg\min_m L_m = \sum_{y \in \mathcal{Y}} (D(y \odot m(y)) - 1)^2$$

(11)

Differently from CycleGAN (Zhu et al., 2017) and DiscoGAN (Kim et al., 2017), AM is not bidirectional and cannot use cycle constraints. We have found that adding magnitude prior $L_1(m(y), 1)$ improves performance and helps prevent collapse. To partially alleviate mode collapse, we use Spectral Norm (Miyato et al., 2018) on the discriminator.

We evaluated our proposed methods:

*GLO Masking (GLOM):* GLO Masking on mel-spectrograms or images at $64 \times 64$ resolution.

*Neural Egg Separation (NES):* The NES method detailed in Sec. 3. Initializing $\mathcal{X}$ estimates using a constant $(0.5)$ mask over $\mathcal{Y}$ training samples.

*Finetuning (FT):* Initializing NES with the $\mathcal{X}$ estimates obtained by GLO Masking.

To upper bound the performance of our method, we also compute a *fully supervised* baseline, for which paired data of $b_i \in \mathcal{B}$, $x_i \in \mathcal{X}$ and $y_i \in \mathcal{Y}$ are available. We train a masking function with the same architecture as used by all other regression methods to directly regress synthetic mixtures to unmixed sources. This method uses more supervision than our method and is an upper bound.

Table 1: Image Separation Accuracy (PSNR dB/SSIM)

| $\mathcal{X}$ | $\mathcal{B}$ | *Const* | *NMF* | *AM* | *GLOM* | *NES* | *FT* | *Supervised* |
|---|---|---|---|---|---|---|---|---|
| 0-4 | 5-9 | 10.6/0.65 | 16.5/0.71 | 17.8/0.83 | 15.1/0.76 | 23.4/**0.95** | **23.9/0.95** | 24.1/0.96 |
| 5-9 | 0-4 | 10.8/0.65 | 15.5/0.66 | 18.2/0.84 | 15.3/0.79 | 23.4/**0.95** | **23.8/0.95** | 24.4/0.96 |
| Bags | Shoes | 6.9/0.48 | 13.9/0.48 | 15.5/0.67 | 15.1/0.66 | 22.3/0.85 | **22.7/0.86** | 22.9/0.86 |
| Shoes | Bags | 10.8/0.65 | 11.8/0.51 | 16.2/0.65 | 14.8/0.65 | 22.4/0.85 | **22.8/0.86** | 22.8/0.86 |

More implementation details can be found in appendix A.

## 4.1 SEPARATING MIXED IMAGES

In this section we evaluate the effectiveness of our method on image mixtures. We conduct experiments both on the simpler MNIST dataset and more complex Shoes and Handbags datasets.

### 4.1.1 MNIST

To evaluate the quality of our method on image separation, we design the following experimental protocol. We split the MNIST dataset (LeCun & Cortes, 2010) into two classes, the first consisting of the digits 0-4 and the second consisting of the digits 5-9. We conduct experiments where one source has an observed distribution $\mathcal{B}$ while the other source has an unobserved distribution $\mathcal{X}$. We use $12k$ $\mathcal{B}$ training images as the $\mathcal{B}$ training set, while for each of the other $12k$ $\mathcal{B}$ training images, we randomly sample a $\mathcal{X}$ image and additively combine the images to create the $\mathcal{Y}$ training set. We evaluate the performance of our method on $5000$ $\mathcal{Y}$ images similarly created from the test set of $\mathcal{X}$ and $\mathcal{B}$. The experiment was repeated for both directions i.e. 0-4 being $\mathcal{B}$ while 5-9 in $\mathcal{X}$, as well as 0-4 being $\mathcal{X}$ while 5-9 in $\mathcal{B}$.

In Tab. 1, we report our results on this task. For each experiment, the top row presents the results (PSNR and SSIM) on the $\mathcal{X}$ test set. Due to the simplicity of the dataset, NMF achieved reasonable performance on this dataset. GLOM achieves better SSIM but worse PSNR than NMF while AM performed 1-2dB better. NES achieves much stronger performance than all other methods, achieving about 1dB worse than the fully supervised performance. Initializing NES with the masks obtained by GLOM, results in similar performance to the fully-supervised upper bound. FT from AM (numbers for finetuning from AM were omitted from the tables for clarity, as they were inferior to finetuning from GLOM in all experiments) achieved similar performance ($24.0/0.95$ and $23.8/0.95$) to FT from GLOM.

### 4.1.2 BAGS AND SHOES

In order to evaluate our method on more realistic images, we evaluate on separating mixtures consisting of pairs of images sampled from the Handbags (Zhu et al., 2016) and Shoes (Yu & Grauman, 2014) datasets, which are commonly used for evaluation of conditional image generation methods. To create each $\mathcal{Y}$ mixture image, we randomly sample a shoe image from the Shoes dataset and a handbag image from the Handbags dataset and sum them. For the observed distribution, we sample another $5000$ different images from a single dataset. We evaluate our method both for cases when the $\mathcal{X}$ class is Shoes and when it is Handbags.

From the results in Tab. 1, we can observe that NMF failed to preserve fine details, penalizing its performance metrics. GLOM (which used a VGG perceptual loss) performed much better, due to greater expressiveness. AM performance was similar to GLOM on this task, as the perceptual loss and stability of training of non-adversarial models helped GLOM greatly. NES performed much better than all other methods, even when initialized from a constant mask. Finetuning from GLOM, helped NES achieve stronger performance, nearly identical to the fully-supervised upper bound. It performed better than finetuning from AM (not shown in table) which achieved $22.5/0.85$ and $22.7/0.86$ . Similar conclusions can be drawn from the qualitative comparison in the figure above.

Table 2: Speech Separation Accuracy (PSNR dB)

| $\mathcal{X}$ | $\mathcal{B}$ | Const | NMF | AM | GLOM | NES | FT | Supervised |
|---|---|---|---|---|---|---|---|---|
| Speech | Noise | 0.0 | 2.4 | 5.7 | 3.3 | **7.5** | **7.5** | 8.3 |

## 4.2 Separating Speech and Environmental Noise

Separating environmental noise from speech is a long standing problem in signal processing. Although supervision for both human speech and natural noises can generally be obtained, we use this task as a benchmark to evaluate our method's performance on audio signals where $\mathcal{X}$ and $\mathcal{B}$ are not dependent. This benchmark is a proxy for tasks for which a clean training set of $\mathcal{X}$ sounds cannot be obtained e.g. for animal sounds in the wild, background sounds training without animal noises can easily be obtained, but clean sounds made by the animal with no background sounds are unlikely to be available.

We obtain clean speech segments from the Oxford-BBC Lip Reading in the Wild (LRW) Dataset (Chung & Zisserman, 2016), and resample the audio to 16 kHz. Audio segments from ESC-50 (Piczak, 2015), a dataset of environmental audio recordings organized into 50 semantic classes, are used as additive noise. Noisy speech clips are created synthetically by first splitting clean speech into clips with duration of 0.5 seconds, and adding a random noise clip, such that the resulting SNR is zero. We then compute a mel-scale spectrogram with 64 bins, using STFT with window size of 25 ms, hop length of 10 ms, and FFT size of 512, resulting in an input audio feature of $64 \times 64$ scalars. Finally, power-law compression is performed with $p = 0.3$, i.e. $A^{0.3}$, where $A$ is the input audio feature.

From the results in Tab. 2, we can observe that GLOM, performed better than Semi-Supervised NMF by about 1dB better. AM training, performed about 2dB better than GLOM. Due to the independence between the sources in this task, NES performed very well, even when trained from a constant mask initialization. Performance was less than 1dB lower than the fully supervised result (while not requiring any clean speech samples). In this setting due to the strong performance of NES, initializing NES with the speech estimates obtained by GLOM (or AM), did not yield improved performance.

## 4.3 Music Separation

Separating vocal music into singing voice and instrumental music as well as instrumental music and drums has been a standard task for the signal processing community. Here our objective is to understand the behavior of our method in settings where $\mathcal{X}$ and $\mathcal{B}$ are dependent (which makes synthesis by addition of random $\mathcal{X}$ and $\mathcal{B}$ training samples a less accurate approximation).

For this task we use the MUSDB18 Dataset (Rafii et al., 2017), which, for each music track, comprises separate signal streams of the mixture, drums, bass, the rest of the accompaniment, and the vocals. We convert the audio tracks to mono, resample to $20480$ Hz, and then follow the procedure detailed in Sec. 4.2 to obtain input audio features.

From the results in Tab. 3, we can observe that NMF was the worst performer in this setting (as its simple bases do not generalize well between songs). GLOM was able to do much better than NMF and was even competitive with NES on Vocal-Instrumental separation. Due to the dependence between the two sources and low SNR, initialization proved important for NES. Constant initialization NES performed similarly to AM and GLOM. Finetuning NES from GLOM masks performed much better than all other methods and was competitive with the supervised baseline. GLOM was much better than AM initialization (not shown in table) that achieved 0.9 and 2.9.

## 5 Discussion

*GLO vs. Adversarial Masking:* GLO Masking as a stand alone technique usually performed worse than Adversarial Masking. On the other hand, finetuning from GLO masks was far better than finetuning from adversarial masks. We speculate that mode collapse, inherent in adversarial training,

Table 3: Music Separation Accuracy (Median SDR dB)

| $\mathcal{X}$ | $\mathcal{B}$ | Const | NMF | AM | GLOM | NES | FT | Supervised |
|------|------|------|------|------|------|------|------|------|
| Vocals | Instrumental | -3.5 | 0.0 | 0.3 | 0.6 | 0.3 | **2.1** | 2.7 |
| Drums | Instrumental | -3.3 | -0.5 | 1.5 | 0.8 | 1.3 | **3.5** | 3.6 |

makes the adversarial masks a lower bound on the $\mathcal{X}$ source distribution. GLOM can result in models that are too loose (i.e. that also encode samples outside of $\mathcal{X}$). But as an initialization for NES finetuning, it is better to have a model that is too loose than a model which is too tight.

*Supervision Protocol:* Supervision is important for source separation. Completely blind source separation is not well specified and simply using general signal statistics is generally unlikely to yield competitive results. Obtaining full supervision by providing a labeled mask for training mixtures is unrealistic but even synthetic supervision in the form of a large training set of clean samples from each source distribution might be unavailable as some sounds are never observed on their own (e.g. sounds of car wheels). Our setting significantly reduces the required supervision to specifying if a certain sound sample contains or does not contain the unobserved source. Such supervision can be quite easily and inexpensively provided. For further sample efficiency increases, we hypothesize that it would be possible to label only a limited set of examples as containing the target sound and not, and to use this seed dataset to finetune a deep sound classifier to extract more examples from an unlabeled dataset. We leave this investigation to future work.

*Signal-Specific Losses:* To showcase the generality of our method, we chose not to encode task specific constraints. In practical applications of our method however we believe that using signal-specific constraints can increase performance. Examples of such constraints include: repetitiveness of music (Rafii & Pardo, 2011), sparsity of singing voice, smoothness of natural images.

*Non-Adversarial Alternatives:* The good performance of GLOM vs. AM on the vocals separation task, suggests that non-adversarial generative methods may be superior to adversarial methods for separation. This has also been observed in other mapping tasks e.g. the improved performance of NAM (Hoshen & Wolf, 2018) over DCGAN (Radford et al., 2015).

*Convergence of NES:* A perfect signal separation function is a stable global minimum of NES as i) the synthetic mixtures are equal to real mixtures ii) real mixtures are perfectly separated. In all NES experiments (with constant, AM or GLOM initialization), NES converged after no more than 10 iterations, typically to different local minima. It is empirically evident that NES is not guaranteed to converge to a global minimum (although it converges to good local minima). We defer formal convergence analysis of NES to future work.

## 6 CONCLUSIONS

In this paper we proposed a novel method—Neural Egg Separation—for separating mixtures of observed and unobserved distributions. We showed that careful initialization using GLO Masking improves results in challenging cases. Our method achieves much better performance than other methods and was usually competitive with full-supervision.

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

## A    ARCHITECTURES

In this section we give details of the architectures used in our experiments:

### A.1    IMAGE ARCHITECTURES:

**GLOM Generator:** GLOM trains generators for $\mathcal{X}$ and $\mathcal{B}$, each generates an image given a latent vector. The architecture followed the architecture first employed by DCGAN (Radford et al., 2015). We used 64 filters across MNIST and Shoes-Bags experiments. MNIST had one fewer layer, owing to its $32 \times 32$ resolution. Generators were followed by sigmoid layers to ensure outputs within $[0, 1]$.

**GAN Discriminator:** The discriminator used by Adversarial Masking is a DCGAN discriminator with filter dimensionality of 64. SpectralNorm is implemented exactly as described in (Miyato et al., 2018).

**Masking Network:** Adversarial Masking, NES and the fully supervised baseline all use the same masking function architecture. The masking function takes a mixed image and outputs a mask, that when multiplied by the image results in the $\mathcal{B}$ source: $b = y \cdot m(y)$. The architecture is an autoencoder similar to the one used in DiscoGAN (Kim et al., 2017). MNIST has two fewer layers owing to its lower resolution. We used 64 filters on the top and bottom layers, and doubling / halving the filter number before / after the autoencoder mid-way layer.

## A.2 Audio Architectures:

GLOM and AM use the same generator and discriminator architectures respectively for audio as they do for images. They operate on mel-scale spectrogram at $64 \times 64$ resolution.

**Masking Network:** The generator for AM operates on $64 \times 64$ mel-scale audio spectrograms. It consists of 3 convolutional and 3 deconvolutional layers with stride 2 and no pooling. Outputs of convolutional layers are normalized with BatchNorm and rectified with ReLU activation, except for the last layer where sigmoid is used.

In addition to the LSGAN loss, an additional magnitude loss is used, with relative weight of $\lambda = 1$.

NES and the supervised method operate on full linear spectrogram of dimensions $257 \times 64$, without compression. They use the same DiscoGAN architecture, which contains two additional convolutional and deconvolutional layers.

## B NMF Semi-Supervised Separation:

In this section we describe our implementation of the NMF semi-supervised source separation baseline (Smaragdis et al., 2007). NMF trains a decomposition: $B = WZ$ where $W$ are the weights and $Z = [z_1, ..., z_N]$ are the per sample latent codes. Both $W$ and $Z$ are non-negative. Regularization is important for the performance of the method. We follow (Hoyer, 2004; Kim & Park, 2007) and use $L_1$ regularization to ensure sparsity of the weights. The optimization problem therefore becomes:

$$L = \|B, WZ\|_2^2 + \alpha |W1|_1^2 = \|[B; 0], [WZ; \sqrt{\alpha} \cdot W1]\|_2^2 \quad s.t. \quad W, Z \geq 0 \tag{12}$$

This equation is solved iteratively using non-negative least squares (NNLS) with the solver by Kim & Park (2011). The $Z$ iteration solves the following NNLS problem:

$$\arg\min_Z \|B, WZ\|_2^2 \quad s.t. \quad Z \geq 0 \tag{13}$$

The $W$ iteration optimizes the following NNLS problem:

$$L = \|[B^T, 0], [Z^T W^T, \sqrt{\alpha} \cdot W1^T]\|_2^2 \quad s.t. \quad W^T \geq 0 \tag{14}$$

$W$ and $Z$ iterations are optimized until convergence.

Following Smaragdis et al. (2007), we first train sparse NMF for the training $\mathcal{B}$ samples: $B = W^B Z^B$. Using the weights from this stage, we proceed to train another NMF decomposition on the residuals of the mixture:

$$L = \|Y, W^B Z^B + W^X Z^X\| \quad s.t. \quad W^B, W^X, Z^B, Z^X \geq 0 \tag{15}$$

The $W^X$ iteration consists of NNLS optimization:

$$\arg\min_{W^X} \|(Y - W^B Z^B)^T, (Z^X)^T (W^X)^T\| \quad s.t. \quad W^X \geq 0 \tag{16}$$

The $Z$ iteration consists of NNLS optimization of both $Z^B$ and $Z^X$ on the mixture samples:

$$\arg\min_{Z^B, Z^X} \|Y, [W^B; W^X][Z^B; Z^X]\| \quad s.t. \quad Z^B, Z^X \geq 0 \tag{17}$$

In the above we neglected the sparsity constraint for pedagogical reasons. It is implemented exactly as in Eq. 14.

At inference time, the optimization is equivalent to Eq. 17. After inference of $Z^B$ and $Z^X$, our estimated $B$ and $Z$ sources are given by:

$$\tilde{B} = W^B Z^B \qquad \tilde{X} = W^X Z^X \tag{18}$$

In our experiments we used $Z$ dimension of 50 and sparsity $\alpha = 1.0$. The hyper-parameters were determined by performance on a small validation set.

Figure 2: A qualitative comparison of speech and noise mixtures separated by GLO and GLOM, as well as NES after $k$ iterations. NES($k$) denotes NES after $k$ iterations. Note that GLO and GLOM share the same mask since GLOM is generated by the mask computed from GLO.

| Mix | GLO | GLOM | NES(1) | NES(2) | NES(3) | NES(4) | NES(5) | GT |
|-----|-----|------|--------|--------|--------|--------|--------|-----|

# C  FURTHER QUALITATIVE ANALYSIS OF GLOM AND NES

We present a qualitative analysis of the results of GLOM and NES. To understand the quality of generations of GLO and the effect of the masking function, we present in Fig.2 the results of the GLO generations given different mixtures from the Speech dataset. We also show the results after the masking operation described in Eq. 10. It can be observed that GLO captures the general features of the sources, but is not able to exactly capture fine detail. The masking operation in GLOM helps it recover more fine-grained details, and results in much cleaner separations.

We also show in Fig.2 the evolution of NES as a function of iteration for the same examples. NES($k$) denotes the result of NES after $k$ iterations. It can be seen the NES converges quite quickly, and results improve further with increasing iterations. In Fig.3, we can observe the performance of NES on the Speech dataset in terms of SDR as a function of iteration. The results are in line with the qualitative examples presented before, NES converges quickly but makes further gains with increasing iterations.

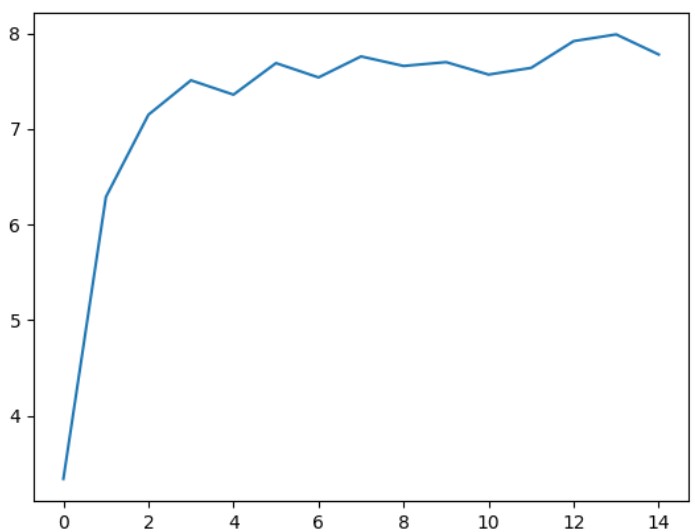

Figure 3: Speech test set separation quality as a function of NES iteration (SDR dB)

