# OpenReview forum: "Neural separation of observed and unobserved distributions"
_ICLR.cc/2019/Conference_

### Official Review · AnonReviewer1 · 2018-11-02
**Approach is reasonable, but insufficient experiment and evidence**

**Rating:** 6
**Confidence:** 4

**Review:**

This paper presents an iterative approach to separate unobserved distribution signal from a mixture with observed distribution. The proposed approach looks reasonable to me, however, the experiment and analysis are insufficient.
1. At test time, does the input also go through the same number of iterations (10)? I would like to see how the separated results evolve over iterations.
2. It is not clear what is the quality of samples generated by GLO. In the image separation task, GLOM performs better than GAN, but worse in other tasks. Analysis is needed here.
3.  I noticed that only in the music separation task, finetuning is significantly better than vanilla NES. Is it because generative models can synthesize more realistic data samples? For example, would the generator learn to synthesize X+B with temporal synchronization? More analysis is also needed here.

============================

I think the reviewer addressed my questions and concerns in the rebuttal, so I raised my rating to 6.

---

> ### Author Response · Authors · 2018-11-06
> **Further analysis**
>
> Thank you for your positive opinion of our method and request for further analysis.
>
> “At test time, does the input also go through the same number of iterations (10)?” : At test time there are no iterations, just a single application of T(), our approach is only iterative in training. The objective of our approach is to create synthetic training samples from the unobserved distribution, which are very similar to unobserved real samples. Once obtained, it suffices to train a separation function T() which minimizes the supervised regression objective - using the synthetic mixtures. This separation function - which is just a single neural network - can be directly applied at test time for any mixture y and yield separated signals x and b (which are given by T(y) and y-T(y)). Following the responses by the reviewers, we significantly simplified the notation and added a description of our method in the form of an algorithm. Our edits are marked in red.
>
> “In the image separation task, GLOM performs better than GAN, but worse in other tasks. Analysis is needed here.” : We would like to explain that we do not claim the generations by GLO are of better quality than GAN. In our experiments, GLOM as a standalone method always underperformed AM in terms of PSNR or SDR, with the sole exception of vocal-instrumental separation. GLOM however does not suffer from mode-dropping, making it more suitable for initializing NES. It is crucial for an initialization to not be too close to a bad local minimum. As GANs generate high quality samples but suffer from mode dropping, they push NES towards suboptimal solutions. We think that GLOM provides an initialization that is further away from bad local minima with missing modes and is therefore a better initialization for NES.
>
> “only in the music separation task, finetuning is significantly better than vanilla NES. Is it because generative models can synthesize more realistic data samples?” : We think that the good performance of GLOM initialization for music separation comes from the fact that it makes no assumption of the relation between the two sources. Vanilla NES assumes independence between the sources, which is not true for instrumental and vocal (or drums) sources in music. It is therefore empirically found to be important to use a good initialization by a technique that does not make the independence assumption (which can be AM or GLOM). We give possible reasons above for GLOM being a better initializer than AM.
>
> “would the generator learn to synthesize X+B with temporal synchronization?” :  It should be noted that GLOM does not consist of a single generator but two generators. G_X() for X and G_B() for B. By using clean training samples from B, we train G_B(), while using the mixture samples we can then train G_X(). GLOM never learns about the dependance between X and B. G_X() and G_B() use unrelated latent spaces (and do not rely on dependence or independence assumptions). We therefore do not expect them to generate signals that are synchronized, but do expect them to be effective in all cases of synchronization.
>
> We will add results addressing the other requests made by the reviewer within the next few days. We would be happy to provide the reviewer with any additional information.

---

> > ### Author Response · Authors · 2018-11-07
> > **Run requested experiments**
> >
> > We have now run the additional experiments requested by the reviewer and added the results to Appendix C.
> >
> > “I would like to see how the separated results evolve over iterations” : We added further visual examples of the separated signals by NES as a function of iteration (Fig. 2). It can be seen that the separation quality gradually improves as a function of time. We plotted SDR as a function of iteration number, giving further evidence for the improvement with iterations of NES (Fig.3).
> >
> > “what is the quality of samples generated by GLO” : As the reviewer requested, we added more visual examples of GLO generations before and after masking (GLOM) in Fig.2. It can be seen that the quality of generation is quite high but further processing by the masking operation has significant benefits.
> >
> > To our understanding, this addresses all the requests made by the reviewer. We would be happy to address any further requests for information.

---

### Official Review · AnonReviewer3 · 2018-11-04
**Well-motivated problem, but the presentation is unclear.**

**Rating:** 5
**Confidence:** 4

**Review:**

This paper describes a signal separation method called neural egg separation (NES).
The separation problem is tackled in a semi-supervised setting where the observed mixture contains a target signal and a background noise, with access to the distributions of target and mixture signals.

The strength of the paper is that it describes the importance of the problem setup for practical use with some motivating examples.
However, some unclear notations weaken the claim of the paper.

Specific comments follow.
* The loss in (1) is unclear.
Assuming latex grammar, \| \| is usually used to denote a vector norm, but (1) has two values inside.
I would write \ell(T(y_i), b_i) to show a loss function, instead of the \| \| style.
More importantly, the loss should be explicitly defined. Does this mean the l2 error?

* The iterative separation process of (2) is even unclear.
Does T^m(b_j + x_i^m) share the parameter of that from previous iterations like T^{m-1}?
Or are the parameters fixed throughout the iterations?

* Use of \cdot.
There may be a confusion between the inner product and element-wise product with the \cdot operator.
Right after (5), there is an inequality z \cdot z \leq 1, which is meant to be the inner product.
On the other hand, the use of \cdot in (8) looks like the element-wise product to describe a masking operation.

Clarifying the objective and overall procedures is necessary for presenting the proposed method.

=================================
EDIT: I confirmed the revisions regarding the notation issues, but there still have confusing parts.
* Definitions of norm operator \| \| is unclear.
  * L_1 is mentioned below (1), and used other parts (3) or Algorithm 1. Equation (12) in Appendix uses |W1|_1^2, which looks like the l1 norm as well. Use consistent notations.
  * Equations (12, 13, 14) uses \|\|_2 or \|\|_1 to specify the type of norm, whereas (5), (6), (7) and other parts after (15) use \|\|. This confuses me. What do you mean by \|\| without subscript?
  * \|\| operator taking to symbols is a weird notation for me. Usually, norm is defined for a single vector (or a matrix). For example in (5), I would write \| b - G(z_b) \|, if you want to measure the difference between b and G(z_b).

The experimental result is impressive, as the other reviewers mention. I strongly recommend clarifying the notation to better deliver the method.

---

> ### Author Response · Authors · 2018-11-06
> **Added suggested edits  and simplified notation**
>
> Thank you for recognizing the importance of our formulation.
>
> “a semi-supervised setting where the observed mixture contains a target signal and a background noise, with access to the distributions of target and mixture signals.” : We would like to highlight that our method is more general than merely separating between target and noise while being given the distribution of the target. We also deal with the much harder case of separating the target being given only pure samples from the nuisance signal (this is the case for the speech and music separation experiments). The only assumption is that one of the sources is given (regardless of which one it is).
>
> Thank you for pointing out some notational improvements:
>
> “The loss in (1) is unclear” : As per the suggestion by the reviewer, we replaced the \|,\| notation by \ell(). We use an L1 loss throughout the paper and made it clear where appropriate.
>
> “Does T^m(b_j + x_i^m) share the parameter of that from previous iterations like T^{m-1}?” : The separation function T^m() is initialized by the weights of the separation function T^{m-1} from the previous iteration (starting from random initialization would also work, however it would require more epochs per iteration). T^m() is of course trained during the iteration. In response to the reviewer’s question, we removed the superscript $m$ altogether, thereby significantly simplifying the notation (while the algorithm does not change).  We also added a description of our method in the form of an algorithm, further improving clarity.
>
> “confusion between the inner product and element-wise product”: We resolved the overloading of dot products with both scalar and element-wise product by replacing all element-wise products by the operation \odot.
>
> We believe this addresses all issues raised by the reviewer. If there are any remaining issues, we would be most enthusiastic to address them.

---

### Official Review · AnonReviewer4 · 2018-11-10
**Interesting ideas with decent empirical results**

**Rating:** 6
**Confidence:** 2

**Review:**

This article presents an interesting if heuristic approach to source separation, NES, buttressed by the use of GLO masking for initialization, with promising results on data generated from synthetic source mixing.

The paper is well written and on the whole clear. My main concern with the work is the empirical nature of the NES iterative procedure. As far as I can tell there is no guarantee of convergence (nor discussion concerning this point). Since i am not familiar with the tasks, it is hard for me to judge the quality of the empirical results -- though the results do seem promising.

re: Bags & shoes task / table 1: "...  Finetuning from GLOM, helped NES achieve stronger performance, nearly identical to the fully-supervised upper bound. It performed better than finetuning from AM (which achieved 22.5/0.85 and 22.7/0.86)": I can't place the first number in the table, therefore i'm not quite sure what is being pointed out here.

re: Music task / table 3: "... GLOM was much better than AM initialization (that achieved 0.9 and 2.9)": I don't see either number in the table. I'd assumed that GLOM was used to fine-tune NES, so I was expecting to see the 2.9 under "FT".

==

I think the authors' response is reasonable. They have added clarifying material to the paper addressing my concerns. I have raised my rating from a 5 to a 6.

---

> ### Author Response · Authors · 2018-11-24
> **Response**
>
> We thank the reviewer for the positive review, particularly for commending the “interesting ideas”, “promising results” and clarity of the paper.
>
> We are able to theoretically show that the correct separation result is a global minimum of NES, but we are yet unaware of a convergence guarantee. The same can be said however for most successful deep learning methods. We establish the good performance of our method via extensive empirical experiments, which the reviewer described as “promising”.  We have added this to the discussion.
>
> Thank you for asking for clarifications on the AM-FT results (different from the AM results). We did not insert them into the table for space considerations, they only appear in the text. We changed the text to elucidate this.
>
> We hope that the reviewer will be able to change the decision to an acceptance given the positive nature of the review.

---

### Author Response · Authors · 2018-12-16
**Thank you for the reviews and increased scores**

We thank the reviewers for the detailed discussion and believe that the manuscript has greatly benefited from it. We also thank all reviewers for increasing their scores.

All reviewers agree that the task we tackle is important, ideas presented are interesting and experimental performance is convincing.

It seems that the only remaining reservations are notational and easy to amend.

---

### Meta-Review · Area_Chair1 · 2018-12-14
**Good formulation, but needs improvement in presentation and experiments**

**Confidence:** 4
**Recommendation:** Reject

**Metareview:**

This paper presents a novel technique for separating signals in a given mixture, a common problem encountered in audio and vision tasks. The algorithm assumes that training samples from only one of the sources and the mixture distributions are available, which is a realistic assumption in a lot of cases. It then iteratively learns a model that can separate the mixture by using the available samples in a clever fashion.

Strengths:
- The novelty lies in how the authors formulate the problem, and the iterative approach used to learn the unknown distribution and thereby improve source separation.
- The use of existing GLO masking techniques for initialization to improve performance is also novel and interesting.

Weaknesses
- There are some concerns around guarantees of convergence. Empirically, the algorithm works well, but it is unclear when the algorithm will fail. Some analysis here would have greatly improved the quality of the paper.
- The reviewers also raised concerns around clarity of presentation and consistency of notation. While the presentation improved after revision, there are parts which remain unclear (e.g., those raised by R3) that may hinder readability and reproducibility.
- The mixing model assumed by the authors is additive, which may not always be the case, e.g. when noise is convolutive (room reverberation, for instance).
- (Minor) Experiments can also be improved. The vision tasks are not very realistic. For the speech separation task, relatively clean speech is easy to obtain. Therefore, it would be worth considering speech as observed, and noise as unobserved. The authors cite separating animal sounds from background, but the task chosen does not quite match that setup.

Overall, the reviewers agree that the paper presents an interesting approach to separation. But given the issues with presentation and evaluations, the recommendation is to reject the paper. We strongly encourage the authors to address these concerns and resubmit in the future.